# Three-Dimensional Culture Models to Study Innate Anti-Tumor Immune Response: Advantages and Disadvantages

**DOI:** 10.3390/cancers13143417

**Published:** 2021-07-08

**Authors:** Alessandro Poggi, Federico Villa, Jordi Leonardo Castrillo Fernadez, Delfina Costa, Maria Raffaella Zocchi, Roberto Benelli

**Affiliations:** 1Molecular Oncology and Angiogenesis Unit, IRCCS Ospedale Policlinico San Martino, 16132 Genoa, Italy; federico.villa@hsanmartino.it (F.V.); leonardo.castrillo@hsanmartino.it (J.L.C.F.); delfina.costa@hsanmartino.it (D.C.); roberto.benelli@hsanmartino.it (R.B.); 2Division on Immunology, Transplants and Infectious Diseases, IRCCS San Raffaele Scientific Institute, 20132 Milan, Italy; zocchi.maria@hsr.it

**Keywords:** spheroids, organoids, alternative culture methods, immune response, innate immunity, NK cells

## Abstract

**Simple Summary:**

The classical approach to study the immune response against a tumor was mixing immune cells with tumor cell suspensions in several experimental settings. These models lack the appropriate tissue architecture in which the immune response takes place and do not consider other cellular and extracellular players of the tumor microenvironment essential to understand the anti-tumor immune response. Thus, to confirm in vitro data, in vivo experiments have been extensively performed, using animal models that may not fully reproduce what happens in humans. Indeed, in animal-based studies, tumors are artificially generated in a short time, and immune cell subsets and receptor-ligands pairs, involved in tumor cells recognition by the immune system, are often different from human counterparts. To reduce the number of animals used, and possibly replace animal models, alternative methods of culture have been developed. Herein, some of these approaches will be described, highlighting their advantages and disadvantages, focusing on natural killer cells as the first line of anti-tumor effector cells able to contrast tumor growth.

**Abstract:**

Several approaches have shown that the immune response against tumors strongly affects patients’ clinical outcome. Thus, the study of anti-tumor immunity is critical to understand and potentiate the mechanisms underlying the elimination of tumor cells. Natural killer (NK) cells are members of innate immunity and represent powerful anti-tumor effectors, able to eliminate tumor cells without a previous sensitization. Thus, the study of their involvement in anti-tumor responses is critical for clinical translation. This analysis has been performed in vitro, co-incubating NK with tumor cells and quantifying the cytotoxic activity of NK cells. In vivo confirmation has been applied to overcome the limits of in vitro testing, however, the innate immunity of mice and humans is different, leading to discrepancies. Different activating receptors on NK cells and counter-ligands on tumor cells are involved in the antitumor response, and innate immunity is strictly dependent on the specific microenvironment where it takes place. Thus, three-dimensional (3D) culture systems, where NK and tumor cells can interact in a tissue-like architecture, have been created. For example, tumor cell spheroids and primary organoids derived from several tumor types, have been used so far to analyze innate immune response, replacing animal models. Herein, we briefly introduce NK cells and analyze and discuss in detail the properties of 3D tumor culture systems and their use for the study of tumor cell interactions with NK cells.

## 1. Introduction

In late 1980s, the seminal findings of Rosenberg and colleagues on the so-called lymphokine activated killer (LAK) cells have shown that LAK-killing of tumor cells can eliminate both autologous and heterologous tumor cells in vitro, and cure mice from melanoma [1,2,3,4]. The transfer to the clinic of Rosenberg’s findings by the systemic administration of interleukin 2 (IL2) showed several drawbacks, such as the capillary leakage syndrome [5,6] leading to fatal outcome in some patients [6]. Indeed, IL2, essential for the generation of LAK cells, gave rise to relevant, unpredictable adverse effects in humans, not affecting murine models [1,2,3,4,5,6]. More recently, the key role of the immune response became evident testing immune-checkpoint (IC) blockers (B) to reactivate the anti-tumor immune response in host bearing tumors [7,8,9,10,11]. In this case, using appropriate tools such as humanized monoclonal antibodies (hmAb) to programmed cell death receptor 1 (PD1), programmed cell death receptor ligand 1 (PDL1) or cytotoxic activated T lymphocyte 4 receptor (CTLA4), it is possible to reactivate the adaptive anti-tumor-specific immune response [7,8,9,10,11]. This strategy is effective when IC-inhibited tumor-specific T cells are already present in the host, thus targeted hmAb can relieve the tumor microenvironment (TME)-mediated immunosuppression [11,12,13,14,15,16].

The study of the molecular mechanisms underlying IC-immunosuppression and patient-specific immune response is difficult in animal models [15,16]. Humanized murine models and patient-derived tumor xenografts (PDX) have been extensively used, with some success. However, it is conceivable that the complex cross-talk among the different cellular and extracellular matrix components of TME is not completely and adequately reconstructed in these hybrid animal models. One example among the others is the species-specificity of some fundamental immunomodulatory cytokines [17,18,19,20,21,22,23,24,25,26].

The innate immunity arm of the anti-tumor immune system has become more and more relevant to improve patient’s response to conventional anti-tumor therapies [27,28,29,30,31,32,33]. Unfortunately, innate cells such as natural killer (NK) cells do not display similar phenotypic and functional features in mice and humans [27,28,29]. To better understand how innate cells can be used to fight cancer, suitable and feasible 3D culture models composed of tumor cells, tumor stromal cells and immune effectors have been set up and used to evaluate the anti-tumor effect of NK cells.

## 2. Developing 3D Culture Models

The addition of appropriate scaffolds and flow-based systems could mimic the architecture of the tumor tissue and the dynamic conditions faced by immune cells approaching and invading the tumor [15,16]. It is evident that the molecular events detected in conventional culture systems, consisting of a mixture of different cell types, cannot be compared to what happens in a 3D-growing tumor mass. On the other side, human tumor cells injected in mice do not find the micro (cell and matrix component) and macro environment (vascular, lymphatic and nervous systems) in which the original tumor mass developed [15,16].

As a possible alternative, several 3D culture systems have been proposed and validated by the EU Reference Laboratories (EURL-ECVAM), as preclinical models, for the selection of anti-tumor drugs [15,16,25], starting from the simplest model of tumor cell homotypic spheroids, composed of a single cell type, through more complex spheroids with tumor and mesenchymal stromal cells (MSC), such as tumor associated fibroblasts (TAF) [16]. Patient-derived organoids, generated from tumors and usually composed of tumor cells at different stage of differentiation, have been used to study and improve the anti-tumor effect of immune cells [34,35,36,37,38,39,40,41]. Some of these developing systems are summarized in Figure 1.

Herein, we will consider the 3D systems used to study NK-tumor cell cross-talk and the possible improvements of these models to better understand this interaction.

## 3. Natural Killer Cells as Anti-Tumor Effectors

Originally, NK cells have been identified as a lymphocyte subset able to recognize tumor cell targets, without priming [42,43,44,45]. This functional definition has been flanked by the identification of phenotypic features identifying NK cells like lymphocytes expressing CD16, the low affinity receptor for fragment crystallizable region of immunoglobulin G, FcγRIIIA [42,43,44,46,47]. Additionally, the neural cell adhesion molecule (NCAM), called CD56, is a typical marker of peripheral blood (PB) NK cells [46,47]. PBNK cells express CD56 at two detectable levels, the so called CD56^low^ and CD56^high^ NK cells, which display a high and a low anti-tumor potential, respectively [46,47]. Of note, CD56^low^ and CD56^high^ PBNK cells can produce low and high amounts of cytokines such as IFNγ and GM-CSF [46,47]. Thus, based on the expression of CD16 and CD56 surface molecules, different NK cell subsets can be distinguished (see Figure 2).

Roughly, CD16^+^CD56^low^ PBNK cells are potent antitumor effector cells and display a strong intracytoplasmic expression of cytotoxic molecules such as perforins and granzymes, whereas CD16^-^CD56^high^ PBNK cells play a regulatory role due to the production of cytokines [46,47,48]. Of note, tissue-isolated NK cells display a surface phenotype more similar to CD16^-^CD56^high^ than CD16^+^CD56^low^ PBNK cells suggesting that tissue resident (TR), and thus tumor infiltrating NK cells are not prone to kill tumor cells. Thus, NK cells are surface CD3/TCR^-^ with a different level of expression of CD16 and CD56 [49,50]. It is not clear to date whether resident NK cells are derived from PBNK cells and whether the two subsets described are different functional stage of the same NK cell at least in humans [51,52] (see Figure 2). Furthermore, several markers of NK cells are shared with the so-called innate lymphoid cells (ILC) including CD56, CD69, CD161, NKp30, NKp44 and NKp46 (these last three receptors called natural cytotoxic receptor, NCR) [49,50,51,52,53,54,55]. Indeed, NK cells have been considered among the different cells that belong to ILC1 subset of ILC [51,52], although it has been claimed that ILC1 and NK cells derive from different progenitor cells [53,54,55]. These molecules play a key role in NK cell-mediated activation and interaction with target cells (see Section 2).

## 4. Basic Interaction between Natural Killer Cells and Tumor Cells

A plethora of receptors expressed on NK cells, and the interaction with the corresponding ligands on tumor target cells, can lead to two main effect: activation or inhibition of NK cells [56,57,58,59]. These receptors–ligands pairs have been described extensively in several reviews that cover both murine and human NK cell features, and their detailed description is beyond the scope of this review. Recent reports can be consulted for a more comprehensive description of these characteristics [60,61,62,63,64,65,66,67,68]. Briefly, activating or inhibiting receptors can be found on NK cells [56,57,58,59], and their engagement eventually results in the killing or the protection from the killing of the target cell [56,57,58,59]. It is commonly accepted that NK cells do no kill self-healthy cells, because they recognize the self-MHC [56,57,58,59,64]. This recognition is mediated by receptors belonging to the Killer Ig like superfamily (KIR) and C-lectin type family (CLIR), that have been characterized in detail in other reviews [57,58,59]. When self-MHC or the receptors for MHC are covered in vitro by specific mAbs, NK cells can kill several self-cell types including tumor cells [57,58,59]. This killing is mediated by activating NK cell receptors whose engagement triggers intracellular calcium increase and activation of Akt kinase, inducing the release of cytotoxic granules at the NK-target cell interface [69,70,71,72,73]. Physiologically, the NK cell-mediated killing can occur when self-cells do not express MHC-class I, for instance when tumors downregulate these antigens to escape from the T cell-mediated antigen specific killing [74,75,76,77]. Nevertheless, several conflicting reports on the topic underline the complexity of self-recognition by NK cells and its regulation [56,57,58,59,78,79].

NK cells recognize and bind to the tumor target through several receptors [42,44,46,58]. The lymphocyte function associated antigen 1 (LFA1), a protein heterodimer composed of CD11a and CD18 chains, interacts with the intercellular adhesion molecule (ICAM) 1 expressed on target cells. ICAM1 is upregulated by IFNγ, induced by the inflammation that can accompany tumor growth [80,81,82,83,84,85]. Several other molecules, such as DNAM1, NKG2D and NCR, strengthen this binding and trigger the activation of killing [42,44,46,58]. DNAM1 and NKG2D receptors bind the corresponding counter-ligands on target cells, poliovirus receptor (PVR) and nectin 2 [86] or the UL binding proteins (ULBP1–6) and the MHC-related molecule A and B (MICA/B) [86,87,88,89,90,91]. Upon this binding NK cells can kill target cells through mainly two different mechanisms: release of lytic factors and enzymes (perforins and granzymes), and action of death inducing molecules such as FasL and tumor necrosis factor (TNF)α [92,93,94].

At the surface of NK cells are also expressed several receptors that deliver, upon engagement with counter-ligands, an inhibitory signal, blocking cytolysis and cytokine release [95,96,97,98,99,100,101,102,103,104,105,106,107,108,109,110]. These receptors, in addition to those involved in the recognition of self-MHC-class I mentioned above, include the leukocyte associated inhibitory receptor 1 (LAIR1) [95,96,97], some sialic acid-binding immunoglobulin-type lectins (siglecs 7 and 17) [98,99], the T cell immunoreceptor with Ig and ITIM domains (TIGIT) [100,101], the T cell activation increased late expression (Tactile, CD96) [102,103,104], the programmed cell death receptors 1 (PD1) [105,106,107], and the T-cell immunoglobulin and mucin-domain containing−3 (Tim 3) [108,109,110]. Thus, NK cells can kill a tumor target only when positive signals overwhelm negative ones [98]. This balance does not only rely on the expression of activating and inhibiting receptors on NK cells, but also on the expression of the corresponding counter ligands on target cells [95,96,97,98,99,100,101,102,103,104,105,106,107,108,109,110] (Figure 2).

The knowledge of the large majority of the molecular mechanisms involved in NK cell-mediated cytolysis has been derived from experiments in conventional culture systems, using tumor target cell suspensions labelled with radionuclide probes and incubated in V-bottomed well plates [111,112,113,114,115,116]. In these culture conditions, NK cells aggress carcinoma epithelial cells (the large majority of solid tumor tissues) as single cells with a spherical shape. These conditions are not similar to what happens in vivo, where tumor cells of epithelial origin grow attached to extracellular matrix in a compact 3D structure. Furthermore, the labelling with a radioactive probe limits the analysis of NK-tumor interactions to short-time assays, due to the toxicity of the probe itself. Thus, some of the findings found in conventional assays are far from the in vivo situation.

## 5. Tumor Cell Spheroids as a 3D Model to Study Tumor Cell Biology and Tumor-Natural Killer Cross-Talk

To transfer in a clinical setting the powerful anti-tumor immune activity of NK cells reported in vitro, several animal models have been used to confirm in vitro observations [18,19,20,21,22,23,24,25,26]. Unfortunately, these models are not fully appropriate, as human neoplastic cells are usually ectopically injected in immunocompromised animals to avoid xenogeneic reaction [18,19,20,21,22,23,24,25,26]. Furthermore, some side effects of the immunotherapy, such as capillary leakage syndrome, have not been detected in animals [1,2,3,4,5,6]. The implant of patient derived xenograft tissue (PDX) in non-obese diabetic severe combined immunodeficient gamma (NOD-SCIDγ or NSG) mice has been extensively used [18,22,24]. The anti-PDX effects of human immune cells, in these humanized mice, only partially resemble the development of a tumor in humans, as tumor microenvironment is complex and composed of different cellular and extracellular components that cannot be replaced by murine cells [18,19,20,21,22,23,24,25,26]. Recently, it has become evident that the replacement of animal experimentation, whenever possible, can be achieved by using 3D culture systems [15,16]. Thus, tumor cell spheroids (this paragraph) and organoids (next paragraph) have been created to study the biological behavior of tumor and immune cells [34,116,117]. Some of the advantages and disadvantages of the above-mentioned models to study immune response are summarized in Table 1.

Spheroids can be derived from either established cell lines or primary isolated tumor cells [35,118,119,120,121,122,123,124,125,126,127,128]. Usually, spheroids are obtained culturing tumor cells in ultra-low adherent plasticware or by hanging drop methods, forcing tumor cells to aggregate and generate tumor sphere in static or microfluidic systems [15,16,126,127,128,129,130,131,132,133,134,135,136]. The spheroids can be composed of a single (homotypic) or, less frequently, different (heterotypic) cell types [15,16,125,126]. Of note, it has been reported that spheroids can be obtained from patients’ tumor-specimens digestion, and self-immune cells can be challenged with self-spheroids to analyze the effect of different drugs and plan their application into the clinic [15,16,117,119,123]. By this way, the 3R (Reduce–Reuse–Recycle) policy has become more and more followed, trying to replace, reduce and refine the use of animal-based assays [134,135]. Herein, we will describe the experimental data derived from spheroid models where the cross-talk between NK and tumor cells have been analyzed.

### 5.1. NK Cells Interaction with Colorectal Carcinoma (CRC) Cells

#### 5.1.1. CRC Biological Features and CRC Spheroids

Several reports have considered CRC spheroids as a good tool to study in a 3D system the features of tumor cells and their interactions with immune cells [125,126,127,136,137,138]. The interest on CRC is related to the strong clinical relevance of this tumor and the impact of immune infiltration on patient prognosis [139,140,141]. Indeed, CRC is the third cancer diagnosed in males, the second in females and it is the second cause of cancer-related death in USA [142,143,144]. CRC has been subdivided into four groups, the Consensus Molecular Subtypes (CMS), based on clinic, pathologic and molecular profiling data [145]. CRC is a very heterogeneous inter and intra patient disease [142,143,144,145]. Recent advances in treating tumors with ICB, are not effective in most CRC, except for microsatellite instability (MSI)-high cancers, showing high neoantigens load [143]. The studies of CRC biology in immunocompetent animal models are limited [146,147,148]. This prompts to generate models to study patients’ immune response variability and drug sensitivity, and to characterize the interaction of immune cells with 3D CRC-derived spheroids [119,120,125,126,127,136]. Tumor cell spheroids can be obtained applying different culture methods [125,126,127,128,129,130,131,132,133] (Figure 3).

#### 5.1.2. Generation of CRC Spheroids and Molecules Involved in NK Cell Mediated Recognition of Tumor Cells

Usually, tumor cells are seeded onto ultra-low adherent culture wells [125,126,127]: this impairs adhesion of tumor cells to plastic substrate and favors the cell-to-cell aggregation. The size of spheroids can be controlled culturing the same number of cells in each well, and this is allowed when tumor cells are put in very little U or V bottomed wells or in hanging-drop plates [129,130,131,132,133]. Spheroids can be embedded into hydrogel drops that solidify at warm temperature, or by calcium addition, and be used as models of initial tumor growth [149,150,151,152,153,154]. This approach allows the contemporary assay of a large number of similar-sized spheroids, incubated with drugs and/or immune cells [154]. On the other hand, spheroids of different size can be randomly obtained when tumor cells are plated onto larger, flat-bottomed plates [125,126,127]. These different approaches can be used to standardize the results. Standardization is easier, having a lot of replicates of the immune-tumor cell interaction, when NK cells are faced with a spheroid of a defined size. Indeed, standardization is a key point to translate in vitro data to the clinic [155,156,157,158,159]. However, the use of spheroids of different size can characterize the behavior of NK cells when challenged with tumors showing different mass density, or inner hypoxic areas [126].

In this context, it has been shown that the NKG2D receptor plays a key role in the recognition of CRC spheroids derived from either established cell lines such as HT29 and DLD1, as well as primary cell suspension from tumor biopsies [119]. This finding is in line with previous observations obtained in conventional culture systems, supporting the notion that 3D culture can validate data from in vitro experiments in a more physiological microenvironment [119]. In this model, the contemporary targeting of NKG2D ligands MICA/B on tumor cells, and the inhibitory receptor NKG2A on NK cells, led to a synergistic effect in blocking tumor cell growth, even if it expressed the NKG2A ligand HLA-E [119]. The authors also originated patient-derived spheroids, mainly composed of a mixture of TAF (due to the initial 2D condition and the presence of serum) and a minor population of tumor epithelial cells. These spheroids were attacked by TIL (either NK or T cells) upon stimulation with IL15 [119]. The use of self-peripheral blood lymphocytes and self-TIL could aid to select appropriate drugs combinations targeting tumor cell growth without impairing the immune response [119].

#### 5.1.3. Invasion and Killing of CRC Spheroids

Haploidentical allogeneic NK cells are an efficient tool to eliminate hematological cancer cells [160], and they have been evaluated as a therapeutic tool for killing CRC spheroids [120,161]. NK cells activated in vitro by the lymphoblastic R69 cell line, without IL2, can kill CRC spheroids derived from Caco−2, HT29 and HCT116 independently from the expression of PDL1 on tumor cells [162]. An inverse correlation between the effector/target ratio used in cytotoxicity assays against spheroids and NK cell infiltration has been shown [160]. At high E:T ratio allogeneic NK cell can kill CRC cells in spheroids without any apparent infiltration, this observation could be intuitive, as lots of NK cells covering the entire spheroid surface would kill all epithelial cells before being able to attempt any invasion of the disaggregating spheroid. On the other hand, the ability of NK cells to kill or invade a tumor spheroid could be triggered by a different balance of stimuli. In this perspective, it is important to define whether a so-called spheroid effectively displays a spherical 3D structure [126]. Indeed, the microscopic observation of a sphere as well as of a discoid is visualized as a circle in x–y axis. The spherical shape can be assessed by the use of microfluidic systems [126]. By this equipment [126], we characterized the exact shape and measured the mass density of CRC cell line spheroids. Mass density is the ratio between the weight and the volume of a spheroid, and it can be considered a parameter typical for each CRC cell line. Interestingly, this parameter changed during both spheroid infiltration by NK cells and NK cell-mediated killing of tumor cells [126]. These findings indicate that mass density can characterize a tumor cell spheroid, and its evaluation can provide new insights on how tumor cells respond to NK cell aggression. Of course, epithelial tumor spheroids lack many other components of TME, and only the analysis of complex heterotypic spheroids could complete the picture. The achievement of such a model is complex, as many paracrine stimuli would be activated and difficultly monitored. However, it is of note that mass density, and possibly other undefined biomechanical features, could give a deeper characterization of tumor behavior and responsiveness to immunotherapy.

### 5.2. NK Cells Interaction with Other Solid Tumor Spheroids

The cross-talk between NK cells and solid tumor cells has also been assessed using spheroids of neuroblastoma, ovarian, cervical and breast carcinomas [121,123,163,164,165]. Neuroblastoma spheroids have been used to assess NK-cell mediated cytotoxicity upon treatment with nutlin-3a, a small molecule that antagonizes the inhibitory effect of MDM2 on p53. The reactivation of p53 triggered the upregulation of DNAM1 ligands on neuroblastoma cells activating NK cell recognition and killing [165]. NK cells cultured with spheroids of ovarian cancer cells (IGROV1, SKOV3, OVCAR3) can downregulate the expression of DNAM1, the same happens co-culturing ovarian cancer specimens with NK cells [121]. The inhibitory effect mediated by DNAM1 downregulation was enforced by the upregulation of the inhibitory receptor TIGIT that can interact, like DNAM1, with PVR and nectin 2, while the levels of CD96, the third receptor for DNAM1 ligands, remained unaffected. These findings indicate that tumor cell line-derived spheroids are a suitable tool to analyze the response of NK cells. Triple negative breast carcinomas (TNBC, HCC1806 and MDA-MB231cell lines) spheroids have been used to assess the immunotherapeutic effect of the bispecific antibody targeting mesothelin and triggering CD16 by a Fab-like domain (MesobsFab) [164]. MesobsFab triggered NK cell-mediated killing and infiltration of tumor spheroids.

Finally, it has been reported that the expression of NKG2D ligands (NKG2DL) was partially downregulated in spheroids of the human cervix squamous cell carcinoma line SiHa and the cervical epidermoid carcinoma CaSki, compared to classical monolayer culture conditions [123]. This downregulation was due, at least in part, to the shedding of NKG2DL, that were detected in the culture medium. Though, these spheroids were infiltrated by NK cells that efficiently killed tumor cells through the activation of NKG2D-NKG2DL interaction. Indeed, anti-NKG2D-blocking antibodies strongly inhibited NK cell-mediated killing [123].

## 6. Tumor Cell Organoids: A More Reliable 3D Model to Study Tumor Cell Biology and Tumor-NK Cells Interactions?

Tumor organoids can be derived mainly by two procedures: using (a) an air–liquid interface (ALI) method, embedding the tissue sample *en-bloc*, without immersion in culture medium [166,167], or (b) matrix-embedded epithelial cells derived from tissue digestion, and continuously growing by the activity of Wnt and other mitogenic and stem factors [40,168,169] (Figure 3B). These approaches can be used to generate organoids from different tumor specimens [40,166,167,168,169] and both can be used to study immune system–tumor cell interactions [40,166,167,168,169]. The ALI method has been described quite recently, and allows to study tumor infiltrating lymphocyte (TIL) phenotype and functional activities for a discrete period and a few culture passages [166]. In this system, TIL can be expanded in the complex microenvironment of the original tumors [166]. Though, the complex structure of the tissue is progressively lost, along time [166]. Tumor associated fibroblasts are present, as smooth muscle alpha actin (SMA) and vimentin positive cells; this may be due to the non-enzymatically based procedure of generation of these organoids, and the presence of serum in the culture medium [166]. However, also myofibroblasts are progressively lost during culture. Thus, this model should be analyzed in short-term experiments for the evaluation of tumor and stroma patient-specific features. The survival of specific cell populations can be improved by the exogenous addition of growth factors. For example, CD3^+^ T lymphocytes (either CD4 or CD8), CD68^+^ macrophages, NKT cells and NK cells can be maintained for about two months in the presence of exogeneous IL2 [166].

On the other hand, matrix-embedded pure epithelial organoid cultures are generated by the proliferation and partial differentiation of epithelial stem cells, such as Lgr5+ cells. The presence of a very defined culture medium, allows to preserve the epithelial stem cell population by appropriate mitogenic stimuli and inhibition of differentiating signals, though it is not permissive for mesenchymal cell growth [170]. These organoids can be indefinitely expanded in culture and allowed to growth in different sizes. This model has been used to proof the possibility of deriving anti-tumor specific T lymphocytes in vitro, able to recognize and kill self-tumor cells [170]. In this case, while the generation of CD8 T cells was tumor antigen specific, the CD4^+^ T cell subset could respond to matrix mouse components. Thus, this system is more suitable to study cytotoxic T cell immune response, than CD4-mediated help. In the next paragraph, we discuss the reports regarding the interaction between organoids and NK cells, while the studies on T cell-mediated immune response are not considered, being treated in specific reviews [168,169,170,171].

### 6.1. How NK Cells Modify Their Behaviour upon Interaction with Organoids

#### 6.1.1. Co-Culture Conditions of Organoids and Immune Cells

Data on how NK cells interact with organoids are scanty and it is relevant to clarify the exact culture conditions in which the experiments have been performed. Usually, immune cells are cultured in conventional RPMI1640 medium supplemented with fetal calf serum, while organoids should be cultured in chemically defined media, with specific growth factors and supplements (such as epidermal growth factor, EGF, fibroblast growth factor, FGF, Wnt3a, R-spondin1, noggin, B27, N-acetyl cysteine, and chemical inhibitors of the TGF-β and p38 pathways) [168,169,170,171]. It is not well-defined whether these different culture conditions may affect the cross-talk between immune and epithelial cells. Furthermore, immune cells should be supported by the addition of exogeneous IL2 or other immune cell stimulating triggers. It has been reported that unfractionated self-peripheral blood mononuclear cells (PBMC) can be co-cultured for a long time with epithelial cell organoids from CRC or non-small cell lung carcinoma (NSCLC) in the presence of IL2, IFNγ, anti-CD3 and anti-CD28 antibodies coated to plastic for 28 days. Anti-PD1 blocking antibodies were used to limit the inhibition of the immune response mediated by the interaction of T lymphocytes-expressed PD1, with PDL1 on epithelial organoids [168,169,170,171]. Under these complex experimental conditions, it has been shown for the first time to our knowledge, that an efficient T lymphocyte-specific immune response against self-tumor organoids can be elicited. This is the proof of principle that the presence of specific anti-tumor T lymphocytes in a given patient can be assessed in vitro [170].

#### 6.1.2. Culture Requirements of NK Cells

NK cells require only IL2 (or another cytokine such as IL12, or IL7, or IL15, or IL21) to proliferate, differently from T lymphocytes, because they express the βγ chains of IL2 receptor [172,173,174]. These cytokines can enhance the basal cytolytic activity of NK cells and trigger an evident cytotoxic effect against carcinoma cells [172,173,174,175,176]. The expression of PD1 is limited to a minor population of polyclonal activated NK lymphocytes and mainly in selected donors [107,177]. Thus, the addition of anti-PD1 antibodies to a culture of NK cells with organoids is not needed. Peripheral blood NK cells, in pancreatic ductal adenocarcinoma (PDAC) patients, display CD16 and CD57 upregulation, while these markers are downregulated in the scanty NK cells infiltrating the tumor, showing also reduced NKG2D and NKp30 activating receptors expression [178]. The co-culture of PBMC from these patients, with self-organoids, determined a phenotype switch of NK cells, with downregulation of CD16, and low CD57, as observed in PDAC tumors. It is of note, that PDAC patients with a fraction of PB CD56^low^ CD16^-^ NK cells over 1% (the median level of this subset), showed less PDAC recurrence than patients with less than 1%. This would indicate that the TME of organoids co-cultured with self-PB NK cells in vitro could induce the selection of NK cell subsets mimicking patient’s situation. Thus, in vitro 3D systems can mimic pathophysiological conditions, supporting their role for a better understanding of the innate immune response against cancer [178].

#### 6.1.3. Interactions of NK Cells and CRC-Derived Organoids

Patients-derived CRC organoids have been used for preclinical assessment of the chimeric antigen receptor (CAR) NK cells [179]. Using the standardized CAR-NK92 model, it was possible to quantify the CAR-mediated cytotoxicity against CRC organoids. First, it has been shown that the cytolytic activity of the CAR-NK92 cell line was strongly impaired by the presence of nicotinamide in the organoid medium, whereas the depletion of this compound did not affect the viability of organoids. Luciferase/GFP expressing CRC organoids were used as targets for EPCAM-specific CAR-NK92, in nicotinamide-free medium. These co-cultures were dynamically monitored, by confocal microscopy imaging and subsequent automatic analysis of recovered data. The cytotoxicity of CAR-NK92 cells against EPCAM^+^ organoids was quantified evaluating both the decrease of luciferase/GFP fluorescence and the area occupied by organoids [179]. In the same model, it has been shown that engineered EGFRvIII-CAR-NK92 cells can kill transduced EGFRvIII-organoids, but not non-transduced organoid controls. This indicates that organoids can be used as a tool to quantify CAR-NK92 cytotoxicity and that EGFRvIII-CAR-NK cells might be used as a therapeutic tool, as the EGFRvIII form of EGFR is a peculiar feature of several tumors [180,181].

### 6.2. Improvements of 3D Culture Systems and Therapeutic Relevance of Studies in 3D Models

#### 6.2.1. Limitations of Spheroids and Organoids as 3D Culture Models

Both tumor spheroids and tumor organoids have limitations in mimicking the physiological conditions where NK cells interact with tumor cells. First, immune cells should reach a tumor by extravasation, interacting with the endothelial cell layer [182,183,184,185]. Afterwards, NK cells need to degrade the extracellular matrix, to reach tumor tissue [186,187,188]. In some instances, they can encounter mesenchymal stromal cells (MSC) that are in close proximity to tumor cells. MSC themselves can be influenced by the growing tumor and can be activated. MSC proximal to the tumor mass do not show altered markers, compared to healthy MSC, resembling reactive MSC instead of tumor-infiltrating MSC [189,190,191]. Thus, MSC close to the tumor have some features of repairing tissue MSC [189,190,191]. Additionally, NK cells come across with several healthy cells before reaching tumor cells [192,193,194]. It is conceivable that NK cells may interact with a tumor-budding spheroid during the progressive growth of the neoplastic mass [195,196,197,198]. The infiltration of the tumor will be different if NK cells reach the tumor from its periphery, or from a vessel generated inside the tumor mass. In this context, the study of the NK-tumor cell interaction has to consider the dynamic conditions that accompany tumor growth and its vascularization [195,196,197,198]. It is evident that, within a tumor mass, a strong heterogeneity is present due to the differential expression of altered genes (mutations, up and down regulation of genes, chromosomal/copy number alterations) in different cell populations, and the heterogeneity of the metabolic state of these cells [199,200,201,202,203]. The expression of mutated genes can be dependent on the metabolic state and the primary source of energy used by tumor cells, and vice versa, some mutations trigger metabolic responses [204,205,206,207].

#### 6.2.2. Microfluidic Model to Study the NK Cell Distribution in Tumor Spheroids

The influence of metabolism on tumor growth/mutation and the immune response could be tested by the generation of microfluidic models [126,208,209,210,211]. For example, two parallel capillary-like structures layered with endothelial cells have been plated in an extracellular matrix hydrogel; in the middle space, between these capillaries, a breast cancer spheroid was co-plated. This model was used to study a therapeutic antibody and NK cells distributions starting from the vessel lumen [212]. NK cells could sense the tumor, actively approaching and infiltrating the spheroid, apparently better than the passive diffusion of the therapeutic antibody. Indeed, antibodies diffused in the hydrogel, but they initially localized only on the outer surface of the spheroid, next to the artificial vessel. Only at 24 hrs the whole periphery of spheroid was covered by the antibody, though it could not penetrate inside the spheroid. NK cells sensed spheroids through several chemokine receptors and penetrated into the inner portion of the spheroid. These NK cells can kill tumor cells both at the periphery and inside the tumor mass [212]. This model is of great interest because the differential penetration of therapeutic antibody and NK cells in an artificial system that resembles the TME can provide information on how the anti-tumor immune response can be regulated. However, it does not take into consideration the plethora of activating and inhibiting signals that NK cells can receive during their localization, as mentioned in Section 3.

#### 6.2.3. Heterotypic Spheroids to Better Mimic TME

The use of heterotypic spheroids would be essential to better understand the complex interaction between anti-tumor effector cells and TME [122,125,213]. Recently, several reports have demonstrated the possibility to reconstitute in vitro the conditions to generate heterotypic spheroid [122,125,213]. The combination of carcinoma cells, fibroblasts and immune cells was used to create 3D heterotypic spheroids with the hanging drop method [122]. In particular, the co-culture of the CRC cell line LS174T and fetal fibroblasts MRC-5 led to the generation of spheroids with a highly compact core of fibroblasts surrounded by carcinoma cells [122]. This kind of spheroids were infiltrated by monocytes and to a lesser extent by CD3^+^ T cells. This model was used to assess the localization and functional effects of IgG-IL2v, that is a variant of IL2 linked to IgG, able to interact only with lymphocytes expressing βγ chains of IL2R such as NK, CD8^+^ cytolytic effector cells and CD4^+^ helper T cells, but not regulatory T cells expressing high levels of CD25. IgG-IL2v could increase T cells and CD3^-^CD56^+^CD69^+^ NK cells infiltration of spheroids, and the killing of the peripheral layer of epithelial carcinoma cells. In this model, the use of T cell bispecific [TCB) antibodies, targeting carcinoembryonic antigen (CEA) or fibroblast activating protein (FAP), showed that T cells could be selectively activated to localize and kill either carcinoma cells or fibroblasts. This heterotypic model could be suitable to assess the efficiency of different therapeutic tools [123].

### 6.3. Future Perspectives and Applications of NK-Tumor Cell Interaction in 3D Models

As reported in the previous paragraphs, 3D models are a suitable tool to dissect how patient-derived NK cells may influence the growth of a tumor mass (Figure 4).

Tumor spheroids and organoids allow the study of the molecular mechanisms mediating lymphocyte and NK cell infiltration of a tumor mass [126]. During this process, it is relevant to analyze the immune synapse between activating and/or inhibiting NK cells receptors and their counter-ligands on tumor cells. NK cell cytoskeleton rearrangements could be analyzed to define if the simultaneous interaction with more than one single tumor cell may influence the redistribution of the cytotoxic granules. During the infiltration of spheroids from the periphery to the center, NK cells probably activate/deactivate different molecular mechanisms allowing migration, protein digestion and matrix/cell interactions. These mechanisms do not play a role in conventional 2D or cell suspension-based culture system. This would suggest that 3D culture models could shed new light on the NK cell signal transduction triggered during infiltration.

Another challenge in a 3D culture models is the quantitative analysis of the cytotoxicity of NK cells in tumor cells. This can be achieved evaluating different parameters in tumor cells: caspase activation, increment/reduction of fluorescent probes (passive markers, active substrates, or reporter constructs), and residual living cells quantification [119,121,123,128,160,163,165,166,168,170,171,178,179]. The quantitative analysis of killing has been assessed mainly by image analysis [119,126,128,212,214,215]. This procedure needs appropriate equipment, a long time for the acquisition of large sets of images at different zeta planes of the 3D structure, and the complex informatic analysis with advanced software [214,215]. Furthermore, the accuracy and precision of results depends on the sensitivity of the equipment and the algorithms of the software [214,215]. The development of easier and standardized quantification assays dedicated to these 3D models is a must, as the conventional cytotoxic tests are clearly insufficient to define the behavior of tumor cells during the interaction with effector cells [126,128].

## 7. Conclusions

The immune system can deeply interact with tumor cells in a hostile TME. While the reactivation of the adaptive immunity by ICB is an additional therapeutic tool against cancer, several solid tumors do not show relevant responses to ICB therapy [7,8,9,13,77,98,143,160,177]. NK cells can successfully eliminate tumor cells, but are regulated by a complex balance of positive and negative signals, delivered from the receptor ligands expressed on tumor cells. The role of MHC class I molecules in shaping NK cell self-reactivity is essential. Thus, to fully understand NK cell interaction with self-tumor cells, appropriate culture models should be developed. 3D culture systems, such as spheroids and organoids derived from patients’ specimens, would allow a deeper analysis of the complex structural architecture of NK-tumor cell recognition/killing. Moreover, they can represent a tool to test therapeutic approaches and novel drugs in an autologous setting. These dynamic models, associated with microfluidic chips, can give new insights on the phenotypic and functional features of NK cells, and their application in cancer therapy.

## Figures and Tables

**Figure 1 cancers-13-03417-f001:**
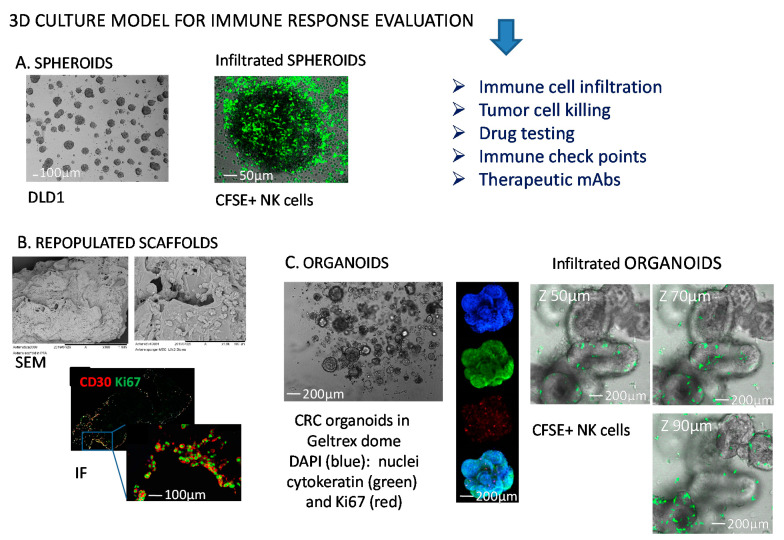
Representative 3D models to study NK-tumor cell interactions. (**A**) Colorectal cancer (CRC) cell spheroids from DLD1 CRC cell line (left) infiltrated with NK cells (right) labelled with the green fluorescence probe carboxy fluorescein succinimidyl ester (CFSE). (**B**) Collagen scaffolds repopulated with mesenchymal stromal cells and Hodgkin’s lymphoma cells (scanning electron microscopy, SEM, upper) or analyzed by immunofluorescence (IF, lower) for the expression of CD30 antigen (HL cells, red) and Ki67 (proliferating antigen, green); (**C**) CRC organoids in a Geltrex^TM^ dome (left), or labelled (middle) with 4′,6-diamidino-2-phenylindole (DAPI, blue), cytokeratin 2 (green), Ki67 (red), or merge (lower image); organoids analyzed by confocal microscopy upon NK cell (green labeled with CFSE) infiltration at different Z-stages (right). The dimension bars are shown in each panel.

**Figure 2 cancers-13-03417-f002:**
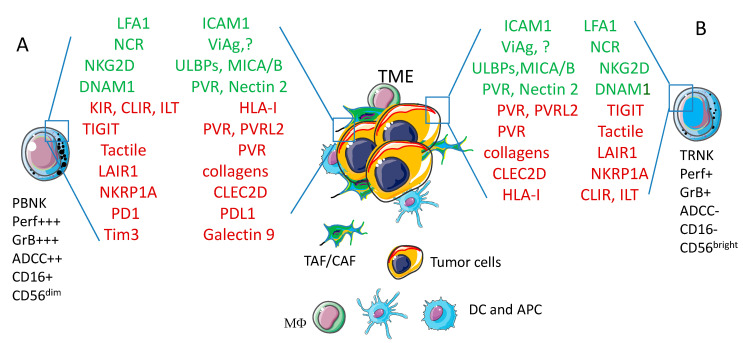
Phenotype of peripheral blood (PBNK) and tissue resident (TRNK) NK cells. The large majority of PBNK (**A**) express CD16 and through this receptor for the Fc portion of immunoglobulins can activate the ADCC that induces the release of perforin and granzymes to kill target cells. A minority of PBNK does not express CD16, but displays high surface amounts of CD56 (CD56^bright^) antigen; these PBNK cells are similar to tissue TRNK (**B**). NK cells express a plethora of activating (green) or inhibiting (red) receptors that can interact with the corresponding ligands expressed on tumor cells within the tumor microenvironment (TME). Other cells of the TME, such as tumor/cancer associated fibroblasts (TAF/CAF), dendritic cells (DC) or antigen presenting cells (APC) and macrophages (MΦ) are also depicted. The engagement of activating receptors triggers cytolysis of tumor cells and/or the release of cytokines while the interaction of inhibiting receptors with corresponding ligands impair NK cell-mediated activities. The final net effect (triggering or inhibition) is determined by the balance of these positive and negative signals. The receptors and ligands mentioned in the figure are analyzed in detail in the text. Acronyms. LFA1: lymphocyte function antigen 1; ICAM1: intercellular cell adhesion molecule 1; NCR: Natural Cytotoxicity Receptor; ViAg: viral antigen; ?: not yet identified ligands; NKG2D: NK related group 2D; ULBPs: UL binding proteins; MICA/B: MHC-related molecule A/B; DNAM1: DNAX adhesion molecule 1; PVR: polio virus receptor; KIR: killer immunoglobulin receptor; CLIR: C-lectin inhibitory receptor; ILT: Immune lymphocyte transcript; HLA-I: human histocompatibility antigen; TIGIT: T cell immunoreceptor with Ig and ITIM domains; PVRL2: poliovirus receptor related 2; LAIR1: Leukocyte associated inhibitory receptor 1; NKRP1A: NK related protein 1A; CLEC2D: C-type lectin domain family 2 member D; PD1: programmed death receptor 1; PDL1: programmed death receptor ligand 1; Tim3:T-cell immunoglobulin and mucin-domain containing−3; Per: perforin; GrB: granzyme B; ADCC: antibody dependent cellular cytotoxicity.

**Figure 3 cancers-13-03417-f003:**
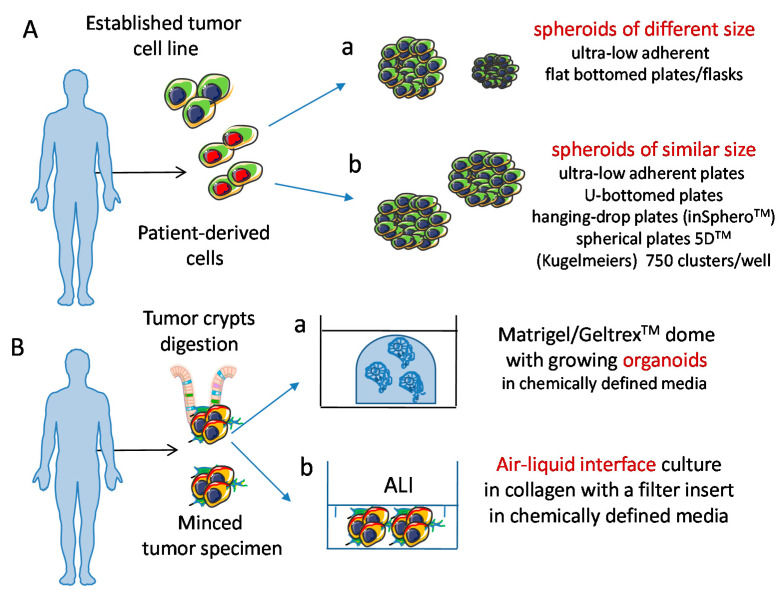
Schematic representation of 3D culture models of tumor cells. (**A**) Spheroids can be obtained from tumor cell lines and tumor cell suspensions. Physical features of spheroids can change, depending on the culture conditions used, resulting in different (a) or similar (b) size. (**B**) Organoids can be derived from tumor cell suspensions seeded either in matrigel/geltrex domes (a) or in air liquid interface (ALI) cultures, (b) and can display a well-defined polarization of molecules like those localized at the basal or apical pole of cells. Importantly, spheroids and organoids can be cultured in chemically defined media; this is of relevance to avoid any influence during the experimental procedures of animal/viral derived factors; this leads to the generation of a more physiologic microenvironment.

**Figure 4 cancers-13-03417-f004:**
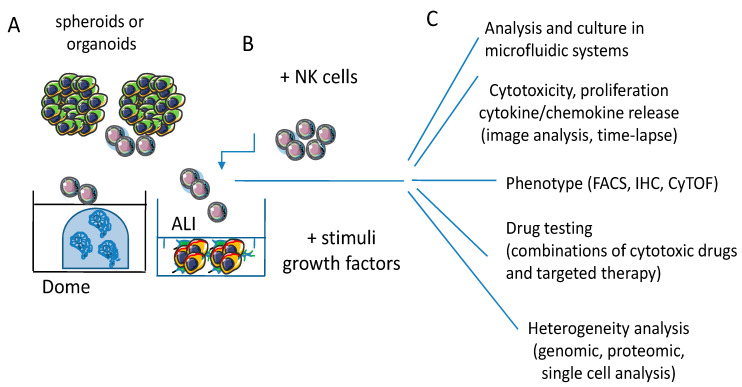
3D culture models for NK/effector-tumor cell interaction. (**A**) Spheroids and organoids allow a more holistic approach to study the interactions between tumor and effector cells, provided the use of chemically defined media with components that do not alter the phenotypic and functional features of epithelial cells and lymphocytes and avoid factors that can metabolically inhibit one or another of these two cell populations. (**B**) The use of specific stimuli such as growth factors or cytokines can mimic in vivo conditions like in conventional cultures. The 3D structure of tumor mass forces the lymphocytes to interact with tumor cells and penetrate into spheroids and/or organoids. This interaction should trigger mechanisms of lymphocyte invasion, while also tumor cells respond to this trigger. (**C**) Endpoints and assays involved in effector lymphocyte/tumor cell 3D cultures analysis. NK/effector cell-mediated cytotoxicity and cytokine release through imaging software tools and time-lapse recording, including computerized microfluidic systems. NK/effector cell phenotype can be assessed using different methodological approaches, such as fluorescence activated cell sorting (FACS), immunohistochemistry (IHC) or mass cytofluorimetry (CyTOF). Drug testing, including combinatorial drug regimens, can be performed evaluating spheroid/organoid size and the differences of metabolism and oxygenation of the tumor mass, as it happens in vivo. The response of tumor cells, as well as that of NK cells, can be analyzed in detail applying genomic and proteomic analysis at the single cell level upon the disaggregation of the 3D culture model.

**Table 1 cancers-13-03417-t001:** Advantages and disadvantages of in vitro and in vivo tumor cell culture models to study interactions with immune cells.

Model Type	Advantages	Disadvantages
**Conventional cultures of established tumor cell lines**	Relative low cost	Limited 3D interaction
	Low care to culture	Limited cell to cell interaction
	Low expertise to culture	Limited cell to matrix interaction
	Easy genetic modification	Limited microenvironment and intercellular communication
	Fast growth	Limited or lack of cellular polarization
	Minimal culture requirement	Genotypic and phenotypic selection of clones after several splitting
	Easy drug testing	Inter and intra laboratories culture selection
	Easy/scalable experimental replicates	Needs frequent authentication
	Easy co-culture experiments with immune cells	
**Patient-derived tumor cell suspension**	Representative of the original tumor immediately after isolation	Difficult genetic stabilization (heterogeneity)
	Ideal for TME and single cell studies	Difficult to culture
		Derived cell lines only partly representing the original tumor
		Low number of cells for functional experiments
**Cell lines-derived or patient-derived spheroids**	Several plasticware tools to get spheroids from single cells	Relative higher cost compared to conventional cultures
	Increment of cell to cell and cell to matrix interactions	Difficulties in getting heterotypic spheroids
	Easier growth quantification compared to organoids	Reduced architectural microenvironment
	Limited culture needs	Difficulties in getting spheroids
	Cultured in well-defined media without serum	Difficulties in setting functional assays
		Difficult experimental standardization
		Need of advanced microscopy equipment for analysis
**Patient-derived organoids**	Partial preservation of cellular interactions and partial polarization	Reciprocal cell interaction and gradient of factors are not always polarized as in vivo
	Genetically engineered	Medium-high care to culture
	Identification of different cell types in the same organoid	High culture cost
	Cultured in well-defined media	Reduced architectural microenvironment
	Partial maintenance of genetic features and heterogeneity	Interaction with stromal components not like in vivo and difficult to set in standard organoid medium
	Culture with self-immune cells	Difficulties in setting functional experiments
		Low-medium frequency of efficient generation from patient
		Difficulties in standardization
		Needs of advanced microscopy equipment for analysis
**Animal models**	Genetically determined	High cost and strong specific skill
	Patient derived xenografts improve study of drug efficacy	Not necessary mirror human cell physiology
	Humanized-mice partly resemble in vivo physiology	The stromal components derive from the animal model
		Difficulties to study immune cell interactions
**Cultures in artificial scaffolds and organ on chip, associated with fluidic systems**	Replaces animal models or reduces the number of animals used	High cost and specific expertise requested
	Resembles more physiological conditions	Difficult to standardize
		Needs new approaches to assess functional activity

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
