# Peer review of "Three-Dimensional Culture Models to Study Innate Anti-Tumor Immune Response: Advantages and Disadvantages"

_cancers, 2021, doi:10.3390/cancers13143417_

Round 1
Reviewer 1 Report
Article entitled “Three dimensional culture models to study innate anti-tumor immune response: advantages and disadvantages” is a timely written review and interesting for the readers in the field.
However, below points to be fixed before accepting this article for the publication.
Comments:
- Figure-1 should be better organized. Big and small letters (example "A" and "a") results in confusion. Diagrammatic representation would be more ideal rather than original data.
- Replace the old references with the latest work.
- Page # 12, line 30, spacing is missing between "couldinduce"
Author Response
Article entitled “Three-dimensional culture models to study innate anti-tumor immune response: advantages and disadvantages” is a timely written review and interesting for the readers in the field.
However, below points to be fixed before accepting this article for the publication.
Comments:
- Figure-1 should be better organized. Big and small letters (example "A" and "a") results in confusion. Diagrammatic representation would be more ideal rather than original data.
We have reorganized figure 1 erasing small letters to facilitate the reader and inserting the dimension bar in each panel according to request of reviewer 3. We decided to use original data to make the reader conscious of how these 3D cultures are in real life.
- Replace the old references with the latest work.
We partially agree with the reviewer, as some old references cite the original article that was the keystone for new insights on the topic. Though, we changed some of these references inserting more recent papers on the topic by the same authors, as follows:
Rosenberg SA. Immersion in the search for effective cancer immunotherapies.Mol Med. 2021 Jun 16;27(1):63. doi: 10.1186/s10020-021-00321-3
Instead of ref number 2
Rosenberg SA.IL-2: the first effective immunotherapy for human cancer.J Immunol. 2014 Jun 15;192(12):5451-8. doi: 10.4049/jimmunol.1490019.
Instead of ref number 3
- Page # 12, line 30, spacing is missing between "couldinduce"
We have corrected this mistake due to some matters of Word program opening the same file with different Windows operating system.
Reviewer 2 Report
The review of Poggi et al. describes three-dimensional tumor cell culture models in the context of their applicability to study anti-tumor NK cell responses. This is an original and interesting paper that may deserve attention of many colleagues involved in cancer research. However, before publication the manuscript requires extensive revision.
- There exist dozens of review papers describing NK cell characteristics and the mechanisms of their interactions with target cells. The respective sections concerning description of NK cells are far too long and should be substantially shortened.
- What is badly missing is a lack of information on the methods used for evaluation of tumor cell killing by NK cells in spheroid/organoid cultures. Authors, please elaborate on how tumor cell cytotoxicity and growth restriction mediated by NK cells can be measured.
- Some paragraphs are too long, and the content is thus hardly digestible.
- Is the last paragraph on page 4 a part of the Figure 2 description?
Author Response
The review of Poggi et al. describes three-dimensional tumor cell culture models in the context of their applicability to study anti-tumor NK cell responses. This is an original and interesting paper that may deserve attention of many colleagues involved in cancer research. However, before publication the manuscript requires extensive revision.
- There exist dozens of review papers describing NK cell characteristics and the mechanisms of their interactions with target cells. The respective sections concerning description of NK cells are far too long and should be substantially shortened.
We have reduced this section leaving essential information on the topic.
- What is badly missing is a lack of information on the methods used for evaluation of tumor cell killing by NK cells in spheroid/organoid cultures. Authors, please elaborate on how tumor cell cytotoxicity and growth restriction mediated by NK cells can be measured.
We have added some information about the different methods used to evaluate the tumor cell killing in 3D culture model as suggested. Indeed, we added this paragraph at the end of the manuscript.
Another challenge in a 3D culture modelsis the quantitative analysis of the cytotoxicity of NK cells in tumor cells. This can be achieved evaluating different parameters in tumor cells:caspase activation, increment/reduction of fluorescent probes (passive markers, active substrates, or reporter constructs), and residual living cells quantification (119,121,123,128,160,163,165,166,168,170,171,178,179). The quantitative analysis of killing has been assessed mainly by image analysis (119,126,128,212,214, 215). This procedure needs appropriate equipments, a long time for the acquisition of large sets of images at different zeta planes of the 3D structure, and the complex informatic analysis with advanced softwares (214,215). Furthermore, the accuracy and precision of results depends on the sensitivity of the equipment and the algorithms of the software (214,215). The development of easier and standardized quantification assays dedicated to these 3D models is a must, as the conventional cytotoxic tests are clearly insufficient to define the behaviour of tumor cells during the interaction with effector cells (126,128).
- Some paragraphs are too long, and the content is thus hardly digestible.
We have inserted some subparagraphs to facilitate the reading.
- Is the last paragraph on page 4 a part of the Figure 2 description?
We have corrected this point. Indeed, some text of figure 2 legend was erroneously included in the main text.
Reviewer 3 Report
This is a very good review highlighting the advantages and disadvantages of three dimensional culture models to study innate anti-tumor immune responses with special focus on NK cells. Only 2 areas need clarifying:
-Are the images original e.g. Figure 1 as if the images have been taken from other articles, a permission is necessary. Moreover, images need scale bars.
-In the last sentence of the introduction, the authors stated that a model contemporat containing tumor cell, TME and immune effctors…..; This sentence should be re-written as we do not add TME. Tumor microenvironment is created through the cell-cell interaction, signaling factors, etc ………
All in all, the manuscript gives an interesting scientific perspective on a field that has recently boomed.
Author Response
This is a very good review highlighting the advantages and disadvantages of three-dimensional culture models to study innate anti-tumor immune responses with special focus on NK cells. Only 2 areas need clarifying:
- Are the images original e.g. Figure 1 as if the images have been taken from other articles, a permission is necessary. Moreover, images need scale bars.
All figures are original images not used for other publications. We added scale bars in each panel except for SEM where the bars were already shown in the first version of our manuscript.
- In the last sentence of the introduction, the authors stated that a model contemporary containing tumor cell, TME and immune effectors...; This sentence should be re-written as we do not add TME. Tumor microenvironment is created through the cell-cell interaction, signaling factors, etc.
We have rewritten the last sentence to make it clearer as follows: “To better understand how innate cells can be used to fight cancer, suitable and feasible 3D culture models composed of tumor cells, tumor stromal cells and immune effectors have been set up and used to evaluate the anti-tumor effect of NK cells.”
All in all, the manuscript gives an interesting scientific perspective on a field that has recently boomed.
Round 2
Reviewer 2 Report
The authors made all required changes and I do not hestitate to recommend publication in a present form.